# Taxonomic and functional components of avian metacommunity structure along an urban gradient

**Erin E. Stukenholtz** [1]* , **Richard D. Stevens** [1,2]

**1** Department of Natural Resources Management, Texas Tech University, Lubbock, TX, United States of America, **2** Natural Science Research Laboratory of the Museum of Texas Tech University, Lubbock, TX, United States of America

☙ These authors contributed equally to this work.
* erin.stukenholtz@ttu.edu

**Data Availability Statement:** The data underlying the results presented in the study are available from Cornell University program eBird. This database is publicly available to anyone and we

## Abstract

Identifying biological processes that structure natural communities has long interested ecologists. Community structure may be determined by various processes, including differential responses of species to environmental characteristics, regional-level spatial influences such as dispersal, or stochasticity generated from ecological drift. Few studies have used the metacommunity paradigm (interacting communities linked by dispersal) to investigate avian community composition along an urban gradient, yet such a theoretical construct may provide insights into species turnover even in unnatural settings such as rural to urban gradients. We measured the influence of spatial and environmental characteristics on two aspects of avian community structure across a gradient of urbanization: 1) taxonomic composition and 2) functional richness based on diet, foraging strategies, nesting locations and morphology. We also measured the relationship between species traits and environmental variables with an RLQ-fourth corner analysis. Together, environmental and spatial processes were significantly related to taxonomic structure and functional richness, but spatial variables accounted for more variation than environmental variables. Fine spatial scales were positively correlated with insectivorous birds and negatively correlated with body and wing size. Urbanization was positively correlated with birds that forage at the canopy level, while emergent wetlands were negatively correlated with birds that nested in cliffs and frugivorous birds. Functional richness and urbanization were significantly related to fine spatial variables. Spatial and environmental factors played an important role in taxonomic and functional structure in avian metacommunity structure. This study highlights the importance of studying multiple aspects of biodiversity, such as taxonomic and functional dimensions, especially when examining effects of complementary spatial and environmental processes.

## Introduction

Community ecology aims to understand the primary mechanisms influencing species abundance and community composition at the local level [1]. Metacommunity theory differentiates

**Funding:** The authors received no specific funding
for this work.

**Competing interests:** The authors have declared
that no competing interests exist.

between local and regional processes that influence community structure [2] and can be simplified into four frameworks: neutral theory, patch-dynamics, species-sorting and mass-effects [2]. Initiated by Hubbell [3], neutral theory assumes that differences among species regarding their niches are nonexistent or unimportant to community structure. Instead, it focuses on randomly fluctuating demographic processes and dispersal limitations and how they influence diversity of local communities [2]. The patch-dynamics framework focuses on how tradeoffs, for example, in competitive and dispersal abilities, influence temporal dynamics of communities in environmentally homogenous landscapes [2]. The species-sorting framework focuses on species responses to a heterogeneous landscape, whereby patterns of presence and absence or even abundance of species reflect selection for suitable habitats [2]. Building upon the metapopulation framework of sources and sinks, the mass-effects framework focuses on how high rates of dispersal of individuals of multiple species into less-suitable habitats facilitate coexistence at the local level due to differing competitive abilities in less suitable patch types [2, 4]. The four metacommunity paradigms focus on species dispersal capabilities, niche similarities and environmental filtering.

Environmental filtering, a component of the species-sorting and mass-effects frameworks, can strongly influence species coexistence [4]. However, environmental characteristics often explain no more than 50% of the variation in taxonomic diversity at the local level [4]. Although spatial processes or stochasticity often strongly characterize the taxonomic composition of communities [4], environmental filtering can have strong influences on related functional traits [4–6], characteristics that are relevant to the performance of an organism are expected to have a strong association with environmental variables [4]. By examining a variety of functional traits within a large species pool, it may be possible to gain a deeper understanding of regional distributions and abundances [4].

Focusing on taxonomic and functional characteristics at local scales has led to variable insights and a lack of information on patterns at regional scales [7]. At local scales, increasing urbanization can adversely affect species presence [8, 9]. In anthropogenically modified landscapes, especially those that are urbanized, there often is increased ambient temperature, fragmentation, and pollution of light, chemicals and noise [8, 10]. In addition, predation on nest sites is often higher [11–13], insecticide use can decrease food abundance for insectivores, and food supplemented by humans can increase resources for granivorous birds [14]. For all these reasons, urban communities tend to be composed primarily of introduced invasive species or highly adaptable native species [15].

Humans are continuously transforming the landscape. Therefore, it is necessary to uncover correlations between species functional traits and regional and local processes to provide a better understanding of effects of anthropogenic modifications on the contemporary biota [4]. Avian community composition has frequently been examined as a response to urban gradients at local scales [16–18]. With the addition of regional processes (i.e., connectivity), we may be able to gain a better understanding of how environmental filtering, habitat selection or dispersal may affect urban community structure to an equal or greater extent than in natural communities [8, 16]. Based on patterns described by other urban studies [9–19], we made the following predictions about environmental variables: 1) birds that are granivorous, nest off the ground, or are invasive will be more associated with urban environments; 2) birds that are native, insectivorous, or ground nesters will be more associated with natural environments; and 3) native species richness will be lower in areas of high urbanization and will increase in more natural or rural areas. These predictions focus primarily on the local scale and do not consider spatial processes. We examined relationships between taxonomy, traits, and environmental variables with spatial structures (fine to coarse spatial scales) across all sampling locations. Because urban gradients (and other land-cover characteristics) may reflect fine spatial

scales for birds that are highly mobile [20], we predict that functional characteristics related to urbanization (such as being granivorous, nesting off the ground, or invasive) will have a higher association with fine spatial scales.

## Materials and methods

### Study area

This study focused on avian communities in Texas, the state with the greatest number of recorded bird species (647) in the United States [21]. Many areas of Texas are highly urbanized. In 2019, Houston (human population: 2,303,482), San Antonio (1,492,510), and Dallas (1,317,929) represented metroplexes with three of the ten largest human populations in the United States. Besides being highly urbanized in parts of the state, Texas is composed of multiple ecoregions. Eastern Texas is characterized by a gradient of pine forests to coastal prairies with wetlands in the south, whereas central Texas has a gradient of cross timbers to open grasslands. Continuing westward there are increases in mesquite, prairies, hill country, canyons and deserts. In the state, precipitation increases from west to east, and temperature increases from north to south [22].

### Bird presence-absence–L matrix

We collected data from the Breeding Bird Survey (https://www.pwrc.usgs.gov/bbs/results/) and eBird (http://ebird.org/ebird/data/download) from 1 May to 31 August 2013 through 2017. Since seasonality can affect species composition, we focused only on species observations made during the breeding season *sensu lato* in Texas. The Breeding Bird Survey was developed by the US Geological Survey's (USGS) Patuxent Wildlife Research Center and Environment Canada's Canadian Wildlife Service to monitor North American bird populations. The eBird database is a citizen science project whose data have demonstrated to be effective for describing patterns of diversity at multiple spatial scales and to be comparable to more standardized data sets [23]. To limit the inherent biases associated with eBird data, we followed methods of Callaghan et al. [24] and Ramesh et al. [25]. We removed the following kinds of checklists: (1) those that did not report all observed species, (2) those that were duplicates from multiple observers who participated in the same sampling event, or (3) those where the observer traveled greater than 5 km or covered more than 500 ha so as to stay within our sampling locations. We retained checklists (4) that had a duration between 5 to 240 minutes and (5) followed traveling, random or stationary protocols. We focused on observations of Passeriformes (215 species), Columbiformes (9 species), and Psittaciformes (2 species). We included orders Columbiformes and Psittaciformes to discern how environmental and spatial processes affect invasive species. Observations from avian surveys were plotted in ArcMap 10.7.1 (Esri, Redlands, California, USA).

To collect information on avian communities, we created a grid across Texas using the fishnet tool in ArcGIS. Fishnet creates rectangular cells in a grid with points at the center of each cell. Cell size was set to 40 km by 40 km. For sampling sites, we created a buffer with a 20 km diameter (area = 314 km$^2$) from the centroid of each cell. A 20 km diameter buffer was chosen to represent an area greater than the home range of all studied species. Home range size is related to body size in animals [26, 27]. Studies on avian home ranges are limited, but many passerines within this study have a home range of less than 9 km$^2$ [28–30]. Some of the larger species within the orders Passeriformes, Columbiformes, and Psittaciformes have home ranges ranging from 0.005 km$^2$ (feral rock pigeon, *Columba livia*) [31] to 325 km$^2$ (common grackle, *Quiscalus quiscula*) [32]. The buffer used here encompassed the site of multiple species home ranges, suggesting that multiple populations were present at each site. Sites were larger when

compared to other studies, which can enhance the probability of species detection [33]. The 20-km distance between each site is greater than the home range of each studied species, which limits the possibility of a single individual being included in more than one site, thereby enhancing independence of data points.

We extracted Breeding Bird Survey and eBird GPS points that were within buffers to make up the communities. To ensure that we limited analyses to well-sampled communities, we included communities if they were represented by 50 or more individuals and exhibited an asymptote in species richness based on a rarefaction curve. We conducted rarefaction curves in the Past 3 statistical program [34]. Communities used in this study had a measured richness that was within the 95% confidence interval of rarefaction curves.

## Trait data—Q matrix

For ecological traits, we collected information on diet and percent foraging strategies (semi-qualitative estimates of foraging strategies) from Wilman et al. [35] and morphological measurements and nesting strategies from Oberholser [36, 37] and Ricklefs [38]. We collected data on native status from Oberholser [36, 37], and defined exotic species as birds not indigenous to the continental U.S. whose distribution expanded due to human facilitation. Average body size, dietary characteristics (diet, foraging strategy and bill length) and nest type are related to environmental characteristics of niches [39–41]. Wing lengths of birds are related to energetic costs of flight and facilitate movements across fragmented landscapes [42]. Because dietary variation depends on location and season, we identified dietary guilds using the item that was in the greatest proportion of recorded dietary items, as done by Wilman et al. [35]. Wilman et al. [35] categorized foraging strata as the relative use of different heights such as ground, understory, midhigh, canopy and aerial levels. In cases where morphological measurements for females and males were collected, we averaged means between the sexes. Like dietary guild, we coded nesting strategies as dummy variables (Table 1).

We characterized trait richness using Rao's quadratic entropy [43–45] using the R package "SYNCSA" [46]. Trait information was not available for all species. Therefore, we performed these analyses on 189 species, 17 fewer than the taxonomic analyses. Rao's quadratic entropy measures the difference among traits. Many traditional methods rely on organizing species into groups, instead of quantifying species characteristics [44]. Furthermore, many methods exclude species abundance [44]. We measured functional trait richness for diet, foraging type, nesting location, bill length, wing length and body mass. Furthermore, we calculated species richness of invasive species and native species for each community.

## Spatial and environmental data–R matrix

To measure the relationship between environmental variables and community composition and functional traits, we collected information on land-cover, precipitation and temperature for each community sampled. We used land-cover data from the 2016 USGS National Land Cover Dataset [47]. Using ArcGIS, we aggregated the land-cover types within each buffer and then expressed each land-cover type as a percentage. The most common land-cover type among the communities was shrubland, with an approximate average area of 94.34 km$^2$ ± 24.03. This was followed by pastures (47.38 km$^2$ ± 14.79), grassland (35.37 km$^2$ ± 12.27), croplands (34.76 km$^2$ ± 13.84) and urbanized areas (31.04 km$^2$ ± 12.26). We extracted the of precipitation and temperature from May to August of 2013 to 2017 for each community using PRISM [48] at a resolution of 16 km$^2$. To examine spatial relationships across communities, we extracted projected Universal Transverse Mercator coordinates (WGS 84, Universal Transverse Mercator Zone 14N) for the center of each site via ArcMap.

**Table 1. Functional avian traits.**

| Traits | Categories | Metrics |
|---|---|---|
| Dietary Guild | Insectivorous | 0, 1 |
| | Scavenger | |
| | Frugivorous | |
| | Nectarivorous | |
| | Granivorous | |
| | Herbivorous | |
| Foraging Strategy | Ground | % |
| | Understory | |
| | Midhigh | |
| | Canopy | |
| | Aerial | |
| Morphometrics | Body Mass | Grams |
| | Wing Length | Millimeters |
| | Bill Length | |
| Nesting Strategy | Ground | 0, 1 |
| | Shrub | |
| | Tree, cavity | |
| | Tree, cup | |
| | Building, cliffs | |
| | Parasitic | |
| Status | Native | 0, 1 |
| | Invasive | |

Traits were collected from avian species that were present in communities from the summer of 2013of to 2017.

## Statistical analyses

To examine metacommunity structure, we constructed derived environmental variables with a principal components analysis (PCA) and derived spatial variables with principal coordinates of neighborhood matrices (PCNM). We performed a PCA on highly correlated environmental variables and reduce the number of dimensions. We used the broken stick method to determine which PCs were significant [49].

To examine spatial relationships among communities, we followed the protocol of Borcard and Legendre [50] using packages *vegan* [51] and *adespatial* [52] in R 4.0.3 [53] by: 1) creating a Euclidean distance matrix with the coordinates associated with each community, 2) computing principal coordinates from a truncated distance matrix with a defined threshold of four times the nearest neighbor sampling distance as suggested by Borcard and Legendre [50], 3) testing significance with a canonical correspondence analysis (CCA) with species occurrences as the dependent matrix and all PCNMs as the independent matrix, and 4) assessing significance with forward selection based on a CCA and retaining only eigenvectors with positive eigenvalues that were significant. Significant PCNMs describe the geographical relationship among communities with the use of different spatial scales [50]. Components ranging numerically low to high represent a gradient of fine to coarse spatial scales [50].

Constrained ordination, such as a CCA, partitions variation among multiple groups of explanatory variables [54] and allows for the examination of unique variation related to a particular explanatory group (i.e., environment) after controlling for shared variation with other explanatory groups (i.e., space) [54]. We examined relationships among spatial factors,

environmental factors and species composition with variation partitioning with a CCA to determine: 1) variation in species composition uniquely related to environmental variables based on significant PCs, 2) variation in species composition uniquely related to spatial factors based on significant PCNMs, and 3) variation in species composition related to spatially structured environmental characteristics [55, 56] in Canoco 5 [57]. The dependent matrix was comprised of species presence for 79 communities, and independent matrices were comprised of five significant spatial PCNMs and three significant environmental PCs. Since multiple independent variables were used, we used the adjusted coefficient of determination ($R^2_{adj}$) to estimate effect size. We used a Monte Carlo approach (999 permutations) to determine the significance of unique variation accounted for by environmental and spatial variables.

To examine the unique relationships of trait richness with environmental and spatial factors, we conducted variation partitioning with a redundancy analysis (RDA) [57] to determine: 1) variation in trait richness uniquely related to environmental characteristics based on significant PCs, 2) variation in trait richness uniquely related to spatial factors based on significant PCNMs, and 3) variation in species composition related to spatially structured environmental characteristics [55, 56] in Canoco 5 [57].

To examine associations between taxonomic (species identity) and traits (morphology, diet, foraging strategy, nesting location, and native/invasive status) with spatial and environmental processes in an anthropogenically modified landscape, we used a multivariate technique (RLQ) and pairwise comparisons (fourth-corner analyses) [58–60]. Both analyses are dependent on three matrices—L (species x site matrix), R (environment x site), and Q (species x trait)—but provide different perspectives on the structure of communities [60]. To characterize the structure of explanatory variables, we conducted a principal components analysis on the site by environment matrix (quantitative data) (R). To characterize trait structure, we conducted a Hill-Smith ordination, that considers categorical data in a species by trait matrix (Q) [61]. To examine the correlation between explanatory and trait variables with species presence, we conducted a correspondence analysis on the site by species matrix (L). Matrices Q and R were coupled using an ordination to create linear combinations that were then linked to species presence/absence in matrix L [62]. The fourth-corner analysis tested the relationship between traits and environmental variables for each species separately, whereas an RLQ analysis is a multivariate analysis of the three matrices [60] that examines all species simultaneously. Combining both RLQ and fourth-corner analyses, we tested for global significance (at α = 0.05) by examining two permutation models: Model 2 and Model 4. Model 2 permuted sites to determine if the relationship between species and environment was significant [63]. Model 4 permuted species occurrences to examine if the relationship between species and traits was significant [63]. The results of both models were combined to limit type 1 error as suggested by Dray and Legendre [63]. RLQ and fourth-corner analyses were performed using the R package *ade4* [64].

## Results

### Bird data–L matrix

We obtained data on species composition of 79 well-sampled communities (communities that exhibited an asymptote with a rarefaction curve, Fig 1). There were 205 species present within these communities (S1 Table). Core species (those that were broadly distributed across sites) were mourning doves (*Zenaida macroura*), white-winged doves (*Streptopelia decaocto*) and northern mockingbirds (*Mimus polyglottos*), which were present at most locations (Fig 2). Many invasive species were common across Texas and would also be considered core species: Eurasian collared-dove (*Streptopelia decaocto*; 75% of sites), house sparrow (*Passer domesticus*;

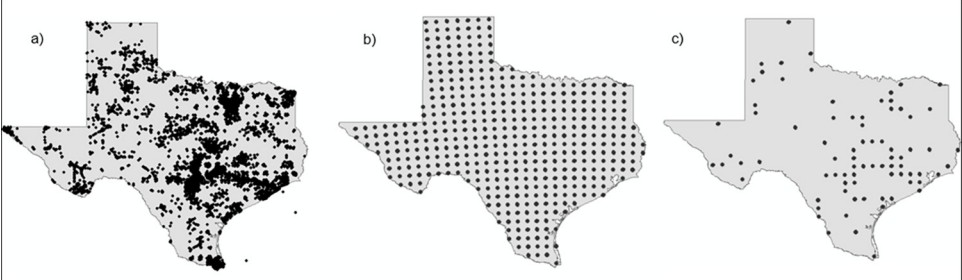

**Fig 1. Community sampling.** Communities were sampled by plotting a) eBird observations from May to August 2013 to 2017. Next, b) a grid was created and in the center of each cell a buffer was created with a 20 km diameter. c) Bird observations were extracted for each buffer, and communities that exhibited an asymptote with a rarefaction curve were kept for analyses.

62%), European starling (*Sturnus vulgari*; 56%) and rock pigeon (*Columba livia*; 44%). The invasive monk parakeet (*Myiopsitta monachus*) was distributed across the fewest sites, being present at only 4%.

## Spatial and land cover data–R matrix

Principal components analysis on 15 land-cover types and average precipitation and temperature yielded three significant PCs (Table 2) that accounted for 55.19% of the variation in environmental characteristics. The first PC accounted for 24.38% of the variation among sites regarding environmental variation and represented a gradient from shrub and grassland to more developed, urban areas (Table 2). The second PC accounted for 17.12% of the variation and encompassed a gradient from areas of high urbanization to pastoral lands, mixed forests and woody wetlands. The last significant PC accounted for 13.69% of the variation and was interpreted as a gradient ranging from forested areas to barren and emergent wetlands.

Constrained spatial analysis using forward selection yielded five significant spatial variables that were retained: PCNM 1 ($R^2_{adj}$ = 2.9%, F = 3.4, p = 0.011), PCNM 4 ($R^2_{adj}$ = 3.1%, F = 3.5,

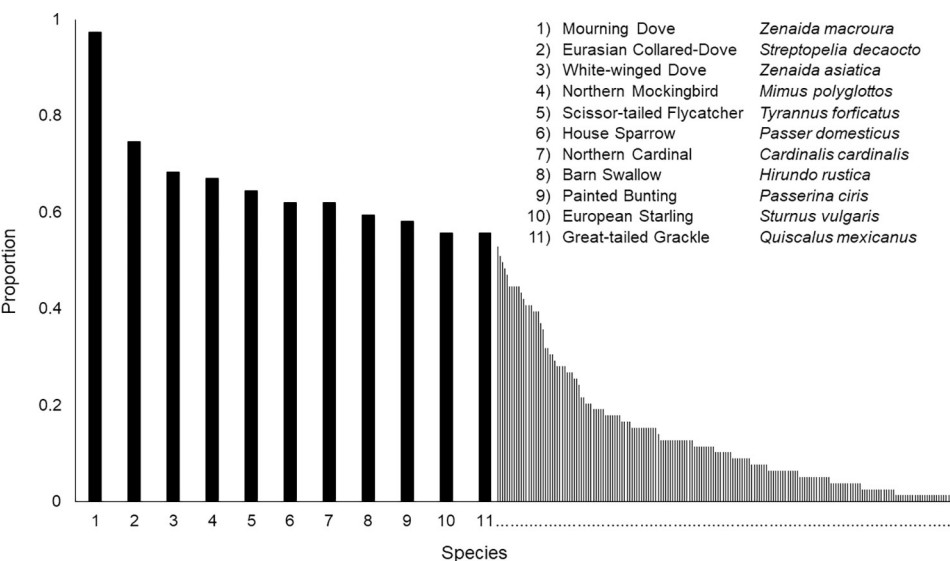

**Fig 2. Species commonality in Texas.** The commonality of avian species throughout the 79 communities during the summer of 2013 to 2017.

**Table 2. Principal components analysis on land-cover types.**

| Variables | Principal Components | | |
|---|---|---|---|
| | 1 | 2 | 3 |
| Barren | 0.13 | 0.18 | **0.65** |
| Cropland | **-0.16** | -0.07 | -0.16 |
| Deciduous Forest | 0.21 | 0.24 | **-0.38** |
| Emergent Wetlands | 0.12 | 0.37 | **0.78** |
| Evergreen Forest | 0.07 | 0.13 | -0.30 |
| Grassland | **-0.20** | -0.32 | -0.19 |
| High Development | **0.83** | **-0.45** | 0.08 |
| Low Development | **0.90** | -0.35 | 0.03 |
| Mid Development | **0.87** | **-0.45** | 0.06 |
| Mixed Forest | 0.21 | **0.58** | **-0.41** |
| Open Development | 0.83 | -0.37 | -0.14 |
| Pasture | 0.22 | **0.68** | **-0.31** |
| Precipitation | 0.64 | 0.48 | -0.20 |
| Shrubland | **-0.61** | **-0.44** | 0.18 |
| Temperature | 0.25 | 0.45 | 0.46 |
| Water | 0.22 | 0.29 | **0.64** |
| Woody Wetland | 0.25 | **0.62** | -0.16 |
| % Variation | 24.38 | 17.12 | 13.69 |

Loadings were collected from principal components that had eigenvalues greater than expected under the broken stick criterion. Derived variables were comprised of land-cover types, precipitation, and temperature characteristics for Texas avian metacommunities from the summer of 2013 to 2017. Bold eigenvalues indicate land-cover types that contribute most to the described gradient.

p = 0.004), PCNM 9 ($R^2_{adj}$ = 1.6%, F = 2.3, p = 0.019), PCNM 14 ($R^2_{adj}$ = 3.0%, F = 3.4, p = 0.008), PCNM 30 ($R^2_{adj}$ = 1.5%, F = 2.3, p = 0.035). The orthogonal variables represented spatial structures that ranged from coarse (PCNM 1) to fine scales (PCNM 30). Coarse spatial scales characterize variation among sites that are the furthest distances from each other, while the finest spatial structures characterize variation among sites that were geographically close.

## Variation partitioning

Environmental variables (land-cover and climatic variables) accounted for a significant amount of unique variation in species composition ($R^2_{adj.}$ = 3.6%, F = 1.9, p = 0.001; Fig 3A), with the first two axes accounting for 2.79% of adjusted variation ($R^2_{adj.}$). The primary axis was a gradient of species composition that was highly correlated with environmental PC 3 (r = 0.82, df = 77, p < 0.001), a gradient of forest to emergent wetlands in the east. The second axis of the CCA was most associated with environmental PC 2 (r = 0.76, df = 77, p < 0.001) and to a lesser extent with environmental PC 1 (r = -0.24, df = 77, p = 0.031). Most species had a higher association with forested land-cover.

Spatial variables accounted for more variation in distribution of species compared to environmental variables ($R^2_{adj.}$ = 4.7%, F = 1.7, p < 0.001; Fig 3B), with the first two axes accounting for 2.77% of adjusted variation in species composition. The first axis was positively correlated with PCNM 1 and PCNM 4 and was negatively associated with PCNM 14 and PCNM 30 (Table 3). Along the first axis, species were correlated with fine to coarse spatial scales (Fig 3B), which indicates that species associated with coarse spatial scales were observed

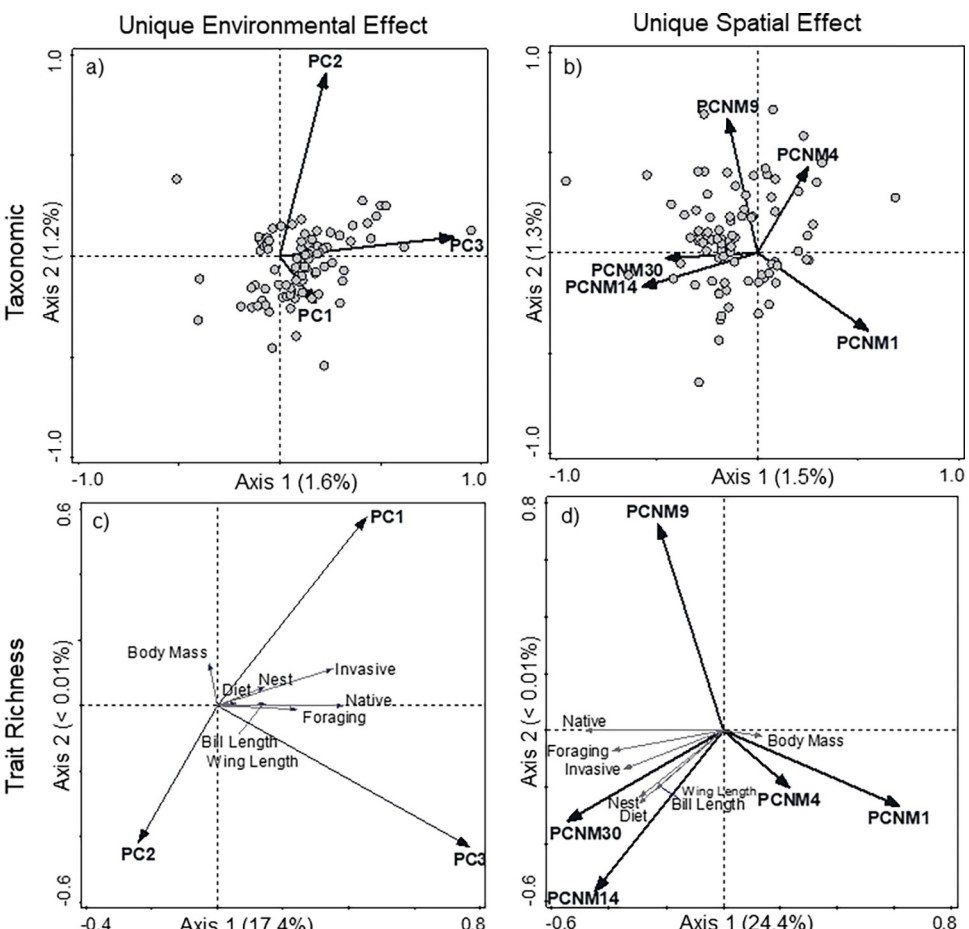

**Fig 3. Relationship between spatial and environmental variables and species occurrences and functional traits.**
Canonical correspondence analysis of species composition with environmental (a) and spatial (b) characteristics as independent variables, and redundancy analysis of trait richness with environmental (c) and spatial (d) characteristics as independent variables for bird species in Texas during the summer of 2013 to 2017. Adjusted variation is represented for each axes. Environmental variables: PC 1- development to shrubland, PC 2- pasture to development, PC 3- emergent wetlands to forests. Spatial variables: PCNM 30- fine spatial scales to PCNM 1- coarse spatial scales.

**Table 3. Correlations of environmental and spatial variables.**

| | | Species Occurrence | | Trait Richness | |
|---|---|---|---|---|---|
| | | **Axis 1** | **Axis 2** | **Axis 1** | **Axis 2** |
| Environmental | PC 1 | 0.19 | **-0.24** | **0.27** | 0.11 |
| | PC 2 | 0.20 | **0.76** | -0.12 | -0.06 |
| | PC 3 | **0.82** | 0.09 | **0.40** | -0.07 |
| Spatial | PCNM 1 | **0.37** | **-0.33** | **0.34** | -0.05 |
| | PCNM 4 | **0.23** | **0.51** | 0.18 | -0.05 |
| | PCNM 9 | -0.10 | **0.59** | -0.14 | 0.15 |
| | PCNM 14 | **-0.38** | -0.15 | **-0.25** | -0.11 |
| | PCNM 30 | **-0.30** | -0.02 | **-0.30** | -0.06 |

Correlations of environmental and spatial variables and axes derived from a Canonical Correspondence Analysis of species occurrences or a Redundancy Analysis of trait richness for birds in Texas during the summer of 2013 to 2017. Bold = P-value less than 0.05. Degrees of freedom for all correlations equal 77.

within sites that had the greatest spatial distance from one another. The second axis of the CCA was positively correlated with PCNM 4 and PCNM 9 and negatively correlated with PCNM 1 (Table 3).

Environmental variables uniquely accounted for a significant amount of variation in trait richness ($R^2_{adj.}$ = 17.4%, F = 6.1, p < 0.001; Fig 3C), with the first axis accounting for 17.4% of adjusted variation ($R^2_{adj.}$). Environmental PCs 1 and 3 were significantly correlated with the first axis of the RDA (Table 3). However, none of the PCs were correlated with the second axis (< 0.001%) of the RDA. All trait richness variables were positively associated with increasing urbanization (PC 1) and emergent wetlands (PC 3; Fig 3C). Spatial variables accounted for more variation in trait richness ($R^2_{adj.}$ = 24.4%, F = 5.8, p < 0.001; Fig 3D) than did environmental variables, with the first axis accounting for 24.4% of the adjusted variation ($R^2_{adj.}$) in trait richness. Spatial variables PCNM 14 and PCNM 30 (fine spatial scales) were negatively correlated with the first axis of the RDA, whereas PCNM 1 (coarse spatial scales) was positively correlated with the first axis of the RDA.

## RLQ and fourth corner analyses

RLQ analysis indicated that there was a significant global relationship between species occurrences and environmental variables (Model 2: p < 0.001). Furthermore, the relationship between species occurrences and traits, while preserving the link between species and environmental variables, was also significant (Model 4: p < 0.001; Fig 4B). This indicated that there was a significant multivariate pattern between traits and environmental variables. Spatial and environmental variables accounted for 91.50% of the variation in trait variables, with the first RLQ axis accounting for 82.26% of the variation in trait richness.

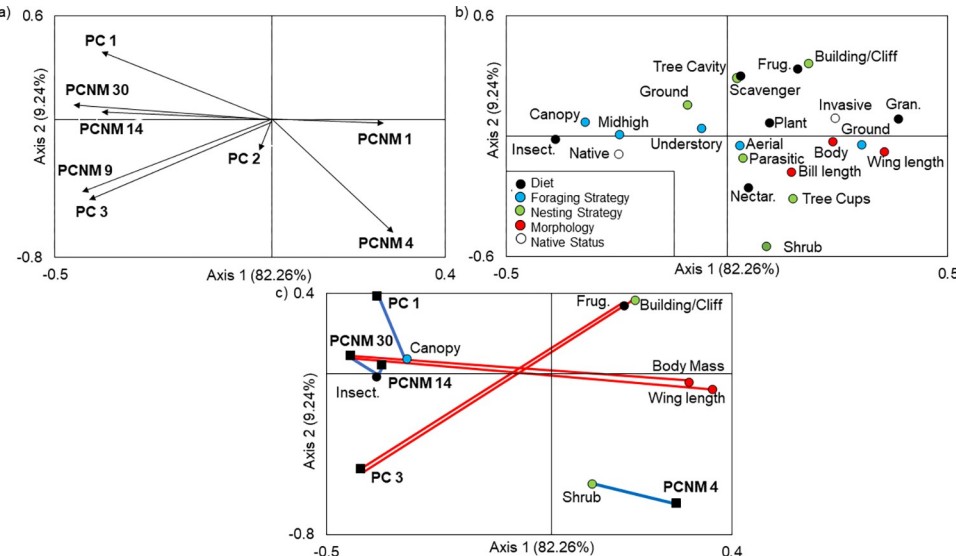

**Fig 4. Sample scores of the first two axes of an RLQ analysis on avian communities in Texas during the summer 2013 to 2017.** RLQ analysis on a) spatial and environmental variables were significantly related to b) avian traits (diet, foraging strategies, nesting locations, morphology and native status. c) Fourth-corner analysis, which measures positive and negative correlations between being explanatory variables and traits, results were added to the RLQ biplot. Variables exhibiting significant positive associations are connected with a blue line and those exhibiting negative associations are connected with a red line. Variables that did not have a significant correlation were removed from biplot c. Environmental variables: PC 1- development to shrubland, PC 2- pasture to development, PC 3- emergent wetlands to forests. Spatial variables: PCNM 30- fine spatial scales to PCNM 1- coarse spatial scales. Diet: Insect. = insectivore, Nectar = nectarivore, Plant = herbivore, Frug. = frugivore, Gran = granivore. Morphology: Body = body mass.

**Table 4. RLQ axes.**

| | | | Axis 1 | Axis 2 |
|---|---|---|---|---|
| Traits | Diet | Insectivorous | -0.15 | -0.02 |
| | | Scavenger | 0.01 | 0.04 |
| | | Frugivorous | 0.06 | 0.04 |
| | | Nectarivorous | 0.02 | -0.03 |
| | | Granivorous | **0.15** | 0.01 |
| | | Plants | 0.04 | 0.01 |
| | Foraging | Ground | 0.12 | -0.01 |
| | | Understory | -0.02 | 0.01 |
| | | Midhigh | **-0.09** | < 0.01 |
| | | Canopy | **-0.13** | 0.01 |
| | | Aerial | 0.01 | -0.01 |
| | Morphometrics | Body Mass | **0.09** | < -0.00 |
| | | Wing Length | **0.14** | -0.01 |
| | | Bill Length | 0.06 | -0.02 |
| | Nesting type | Ground | -0.04 | 0.02 |
| | | Shrub | **0.04** | -0.07 |
| | | Tree Cavity | 0.01 | 0.04 |
| | | Tree Cups | 0.06 | -0.04 |
| | | Building/Cliffsides | 0.01 | 0.04 |
| | | Parasitic | 0.01 | -0.01 |
| | Status | Invasive | **0.10** | 0.01 |
| | | Native | **-0.10** | -0.01 |
| Explanatory | Environmental | PC1 | **-0.09** | 0.05 |
| | | PC 2 | -0.01 | -0.02 |
| | | PC 3 | **-0.01** | -0.06 |
| | Spatial | PCNM 1 | 0.06 | < -0.01 |
| | | PCNM 4 | 0.07 | **-0.07** |
| | | PCNM 9 | **-0.11** | -0.05 |
| | | PCNM 14 | **-0.09** | 0.01 |
| | | PCNM 30 | **-0.11** | 0.01 |

RLQ axes correlations with trait and explanatory variables. Bold = P less than 0.05.

Fourth-corner analysis indicated that 8 out of 176 bivariate associations were significant (Fig 4C). Developed, urban areas (PC 1) were positively associated with foraging at canopy level (r = 0.09, p = 0.038). Birds with a frugivorous diet (r = -0.06, p = 0.038) or that nested on cliffs or buildings (r = -0.07, p = 0.036) were negatively associated with emergent wetlands (PC 33). Birds that nested in shrubs were correlated with coarse spatial scales (PCNM 4: r = 0.07, p = 0.036), indicating that shrub nesters were widely distributed across the communities. Species with an insectivorous diet (PCNM 14: r = 0.10, p = 0.038, PCNM 30: r = 0.10, p = 0.038) and a small body size (PCNM 30: r = -0.070, p = 0.036) and wing length (PCNM 30: r = -0.09, p = 0.038) were significantly associated with fine spatial scales. Insectivorous and smaller birds occupied communities near one another.

There were significant associations among RLQ axes, traits, and explanatory variables when combining fourth-corner and RLQ approaches (Table 4). The first RLQ axis was negatively correlated with increasing urbanization (PC1), emergent wetlands (PC3), and intermediate to fine spatial scales (PCNM 9, PCNM 14 and PCNM 30). For trait variables, RLQ axis 1 was

negatively associated with an insectivorous diet, foraging at mid-high and canopy levels and being native. In contrast, the first axis of the RLQ was positively related to granivorous diets, body mass, wing length, nesting in shrubs and being invasive. Axis 2 was significantly related to coarse spatial scales (PCNM 4) but no other variables.

## Discussion

Birds were highly diverse in their response to environmental gradients and spatial distribution. Spatial structure accounted for more unique variation than environmental characteristics with respect to species composition and trait diversity. Species that were associated with coarse spatial scales were observed at sites that were the farthest apart from each other. Functional trait richness was highly correlated with fine spatial scales potentially indicating that spatially close sites were highly diverse. Spatial and environmental variables also accounted for a large portion of the variation in RLQ matrices. However, environmental variables, specifically urbanization, were not as strongly related to community structure as originally predicted. Canopy foraging was the only trait significantly and positively related to increased urbanization, specifically low development. Other avian metacommunity studies have also detailed the importance of environmental variables (species-sorting framework) on community structure [65–68]. As with other studies, unexplained variation was a major characteristic of metacommunity structure in human-modified landscapes, but environmental filtering and dispersal are often significant contributors [69].

How species respond to environmental changes and community dynamics may have underlying spatial structures [7]. However, few studies add a spatial component to analyses [56]. In a meta-analysis of metacommunities conducted by Cottenie [56], most studies demonstrated that environmental factors were the main driving force of community structure, indicating that species-sorting was the most common framework explaining metacommunity structure. The second most prominent structure was a combination of spatial and environmental variables indicating a combination of mass-effects and species-sorting [56]. Other avian metacommunity studies have also detailed the importance of environmental variables (species-sorting framework) on community structure [65–68]. This study demonstrated a mass-effect framework, with not only environmental characteristics being important, but that spatial structures added significant explanatory value to the model suggesting the importance of dispersal in community assembly.

Principal coordinates of neighborhood matrices represent eigenvector decompositions of spatial scales and sites [55]. Following Borcard et al. classification, PCNMs for this study can be classified into three different spatial groups: coarse (PCNM 1, 4, and 9), intermediate (PCNM 14), and fine-scaled (PCNM 30) [70]. Nesting in shrubs was positively correlated with coarse spatial variation, indicating that shrub nesters were widely dispersed throughout Texas. This result is probably due to the widespread distribution of shrubland and woodlands throughout Texas that provide abundant nesting opportunities. Communities that were near each other in the southeast of Texas had highly correlated traits and species richness. These correlations are potentially rooted in environmental variables that are related to finer spatial scales [71].

Urbanization and emergent wetlands were correlated with fine to intermediate spatial scales within the RLQ analysis, potentially indicating that species are responding to fine spatially structured variables. Urbanization can increase or decrease diversity depending on invasive species introduction, spatial heterogeneity, disturbance or spatial scale [71]. Since urbanization can greatly influence species distribution due to biotic and environmental conditions, we predicted that urbanization would influence species composition and functional traits. However,

urbanization was not significantly related to species composition but was correlated with invasive species richness. Urbanization may greatly influence biodiversity via multiple synergistic effects, such as increasing temperature, water availability, primary productivity, novel biotic interactions with invasive species, etc. [8]. Although these effects can impact multiple native species, many invasive birds thrive in cities and can outcompete similar native species [72–74]. Adaptations, such as behavioral flexibility [74, 75], ecological generalism [75, 76] and human tolerance [75, 76] have all been attributed to the success of invasive species in urban environments which may explain the correlation between invasive species richness and urbanization in this study.

Native species and urbanization exhibited the same association with RLQ axis 1 (Table 4). Although urbanized areas are often characterized by decreased biodiversity compared to surrounding natural habitats [8], there are many positive attributes that can increase native biodiversity in cities, such as higher productivity, resource availability and connectivity [8, 77]. The connection between native species and urbanization may be due to increased primary productivity via urban parks (categorized as open and low development in the NLCD) that can alleviate negative effects of urbanization [8, 78, 79] or harsh environments of arid cities [80]. Parks can increase richness by contributing a wider variety of food and nesting sites [81]. Ground nesting birds often are more prevalent in parks, whereas cavity and tree nesters are often more prevalent in allotment gardens (e.g., community gardens) [82]. Canopy foragers are often prevalent in parks [83], explaining the positive correlation between urbanization and canopy foragers. Examining a multitude of characteristics, such as parks, within cities may be key to understanding how urbanization influences species.

Future studies would benefit from more censuses in the western region of Texas and from considering other kinds of environmental variables. Most of the communities examined herein come from the eastern region of the state, probably due to human population density being greater in the east and thus more people out observing birds. This may indicate why some species were correlated with urbanization and why insectivorous birds were correlated with finer spatial scales. Gathering more information in more isolated locations throughout Texas would provide insights into how environmental and spatial characteristics influence community composition. Moreover, vegetation complexity and quality influence avian distribution [84], and the addition of these variables to the model might improve understanding of metacommunity structure.

When examining urbanization, most studies have primarily focused on local scales, where results can be variable over time. This study demonstrates the importance of spatial variables and spatially structured environmental variables on community structure. There was no correlation between urbanization and some avian characteristics as we predicted. Instead, this study demonstrated that trait and taxonomic richness were correlated with urbanization. Furthermore, urbanization was correlated with fine-scale spatial variation. Therefore, understanding environmental filtering and scale-dependence, especially at fine scales, is essential for understanding species and trait distributions.

## Supporting information

**S1 Fig. Correlation matrix of significant PCAs and PCNMs.**
(TIF)

**S1 Table. Canonical correspondence analysis with avian species.** Canonical correspondence analysis was conducted on species occurrences and environmental and spatial variables in Texas from May to August from 2013 to 2017. Loadings from canonical correspondence analysis for avian species responses to environmental and spatial variables are presented. This table

corresponds to Fig 3A, 3B of the 3C, the first axis of the CCA was positively associated with PC 3, and the second axis to PC2 and to a lesser extent negatively associated with PC 1. For species responses to spatial structures, the first axis of CCA was positively associated with PCNM 1 and negatively associated with PCNM 14 and 30. The second axis was positively associate with PCNM 9 and 14 and negatively associated with PCNM 1. Environmental variables: PC 1- development to shrubland, PC 2- pasture to development, PC 3- emergent wetlands to forests. Spatial variables: PCNM 30- fine spatial scales to PCNM 1- coarse spatial scales.
(DOCX)

**S2 Table. Statistics from variation partitioning.** Statistics from variation partitioning with correspondence analysis to examine the relationship between on avian taxonomy and functional traits and independent variables environmental and spatial variables. Environmental variables represent land-cover and climate data, and spatial variables represent coarse to fine spatial scales. For functional traits, axis 1 was the only significant axis for both environmental and spatial variables. Explained variation is cumulative variation for each axes.
(DOCX)

## Acknowledgments

We would like to thank eBird for the data used in this manuscript, and Macy Krishnamoorthy and Ty Hopper for their edits.

## Author Contributions

**Conceptualization:** Erin E. Stukenholtz, Richard D. Stevens.

**Data curation:** Erin E. Stukenholtz.

**Formal analysis:** Erin E. Stukenholtz.

**Investigation:** Erin E. Stukenholtz.

**Methodology:** Erin E. Stukenholtz.

**Project administration:** Erin E. Stukenholtz, Richard D. Stevens.

**Resources:** Erin E. Stukenholtz.

**Supervision:** Richard D. Stevens.

**Validation:** Richard D. Stevens.

**Writing – original draft:** Erin E. Stukenholtz.

**Writing – review & editing:** Erin E. Stukenholtz, Richard D. Stevens.

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
