## [Decision Letter · Decision Letter 0]

23 Feb 2022

PONE-D-21-38671Taxonomic and functional components of avian metacommunity structurePLOS ONE

Dear Dr. Stukenholtz,

Thank you for submitting your manuscript to PLOS ONE. After careful consideration, we feel that it has merit but does not fully meet PLOS ONE’s publication criteria as it currently stands. Therefore, we invite you to submit a revised version of the manuscript that addresses the points raised during the review process.

 As you will see, both external Reviewers had opposed opinions on your manuscript. For this reason, I also reviewed it. The first Reviewer was very positive and I agree with him that your study is a solid piece of work: it is based on an extensive dataset, and the analyses are based on state-of-the-art methods concerning spatial structures and the relationships between traits and the environment. I however also agree with the second Reviewer that many results are mostly confirmatory (but this is not disqualifying). My main problem with your analyses is that the results are biologically very difficult to decipher. The Discussion section provides a clearer description of the results, but how this description emanates from the results remain obscure. One reason is the use of summary variables for the environment (PC1...). Their use is fine for variation partitioning, but I would stick to the original variables (after selecting the most influential ones) for the Figures. It is also necessary to label at least some species in the figures and to provide tables of their coordinates in the SI. In general, I found the results cryptic with a very dry text. These modifications should make your manuscript easier to read and hopefully more impactful. Please also consider carefully the comments of Reviewer 2, notably concerning the functional traits.

We look forward to receiving your revised manuscript.

Kind regards,

Louis-Felix Bersier, Ph.D.

Academic Editor

PLOS ONE

Journal Requirements:

“There was no funding for this project.”

4. Please upload a copy of Figure 5, to which you refer in your text on page 20. If the figure is no longer to be included as part of the submission please remove all reference to it within the text.

Additional Editor Comments :

You will find my detailed comments in the attached file (PONE-D-21-38671_LFB.pdf)

Reviewers' comments:

Reviewer's Responses to Questions

**Comments to the Author**

1. Is the manuscript technically sound, and do the data support the conclusions?

Reviewer #1: Yes

Reviewer #2: No

2. Has the statistical analysis been performed appropriately and rigorously? 

Reviewer #1: Yes

Reviewer #2: Yes

3. Have the authors made all data underlying the findings in their manuscript fully available?

Reviewer #1: Yes

Reviewer #2: Yes

4. Is the manuscript presented in an intelligible fashion and written in standard English?

Reviewer #1: Yes

Reviewer #2: Yes

5. Review Comments to the Author

Reviewer #1: This interesting study about structure of meta-community. It well written and data were analysed nicely.

I have some minor comments:

Line 46: the average area of urbanization was 31.0 km2. Is this area large or small. Where to compare it?

Lines 1111-113. Add references to your predictions.

Lines 113-115. I did not find the results from this prediction (3). Add references to your prediction. There is lot of previous publications about species richness in relation to urbanization.

Lines 153-154. How intensively birds was counted from 20 km x 20 km squares. How equal bird species counting were in each square?

Lines 159-160. How closely to asymptotic species richness (xx%?). Note that the species richness increasing with bird counting intensity. What program you used to calculate rarefaction. Are rarefaction calculated each 20 km x 20 km square or larger area. Add more details.

Lines 300 - 303. Omit 'Degrees of freedom ...'. It is twice in the table 3 text.

Line 350. Is it really 'individuals'? May be it is 'species'.

Lines 351- 352. Sittidae is twice.

Lines 351-353. Would you add figures from those correlation.

Figure 1. Would you re-draw this figure. See more details: McGeoch & Gaston 2002: Occupancy frequency distributions: patterns, artefacts and mechanisms. Biological Reviews, 77, 311-331. Write also some words to the results and discussion sections. You can also analysed the distribution pattern see more details:

Hui C. (2012) Scale effect and bimodality in the frequency distribution of species occupancy. Community Ecology, 13, 30-35.

Jenkins D.G. (2011) Ranked species occupancy curves reveal common patterns among diverse metacommunities. Global Ecology and Biogeography, 20, 486-497.

Reviewer #2: In this study, Stukenholtz and Stevens explore the influence of spatial and environmental features on taxonomic and community richness of bird communities along an urbanization gradient in Texas (USA). I found the study rather confirmative. Most results are not novel and the contribution and relevance of this study in relation to the literature is not clear.

Some specific comments:

Title: In my humble opinion, this study is not about metacommunities. The title is not very informative and should be modified.

L41 “functional richness of diet” sounds odd. Please, reword.

L45-46 This info is not relevant here.

L50-51 This is a rather vague statement. Please, elaborate a bit.

L51-53 I think the conclusions of this study should be much improved in order to attract the attention of a broad audience.

L111-115 Some of these predictions constitute well-known patterns and someone would argue that rather than hypotheses to be tested, they constitute truisms. Please, specify the main novelty of this study in relation to previous work.

Bird Data: It seems that bird surveys were conducted in different habitats along an environmental gradient. It is known that bird detectability can vary among habitats (e.g., detection probability is higher in open vs. closed habitats). How did you account for detection biases in this study?

Trait Data: The Hand Wing index is a better proxy for dispersal capacity than wing length. This variable can be obtained from recent studies (see e.g., Sheard et al. 2020. Nature Communications).

Trait Data: Foraging variables (%): it is likely these variables are correlated so I think they could be summarized into 1-2 axes by means of a Principal Component Analyses without loss of information.

Trait Data: Body mass (length) and bill length are highly correlated, so I would use the residuals of bill length (i.e., size-corrected bill length) after a phylogenetic size-correction (Revell, 2009) instead of the raw variable.

I wonder why authors do not use other more traditional measures of functional diversity instead of functional richness, a metric that depends on species richness.

L262-265 This info should be given in Material and Methods.

L265-272 This belongs to M&M.

L305 What about the relationship between species richness and environmental variables (e.g., urbanization gradient)? Is species richness significantly associated with trait richness?

Results: This section is too wordy and not easy to read. I think it should be trimmed by half.

L369-379 This paragraph fits better in Introduction.

L381-384 In my opinion, these results are not novel at all. These are rather platitudes.

L384-385 Another quite obvious result. Obviously, trait diversity will be higher in ecosystems where terrestrial and semi-aquatic species coexist.

L390-391 Indeed, trees and cliffs are scarcer in wetlands, so this relationship is quite obvious and lacking of interest.

L404 Traits like brain mass (which is available for a large number of species) would be of interest in this context.

439-441 Conclusions are not conclusive at all and the take-home message is a bit disappointing. Authors should emphasize the main merits of their study.

Fig. 3b: The number of variables is so high that this figure is hardly interpretable.

Fig. 3d: Unclear figure. It is almost impossible to discern among the large number of symbols used to identify each family.

6. PLOS authors have the option to publish the peer review history of their article (what does this mean?). If published, this will include your full peer review and any attached files.

Reviewer #1: No

Reviewer #2: No

---

## [Author Response · Author response to Decision Letter 0]

12 May 2022

Comments from Editor:

We note that you have stated that you will provide repository information for your data at acceptance. Should your manuscript be accepted for publication, we will hold it until you provide the relevant accession numbers or DOIs necessary to access your data. If you wish to make changes to your Data Availability statement, please describe these changes in your cover letter and we will update your Data Availability statement to reflect the information you provide.

Data for this study was collected from a citizen scientist database. Therefore, we will need to change our Data Availability Statement. We will not have data to provide once accepted, since it is already publicly available and does not belong to us. Thank you for helping us to clarify this situation.

Thank you for stating the following financial disclosure:

The authors received no specific funding for this work.

Please upload a copy of Figure 5, to which you refer in your text on page 20. If the figure is no longer to be included as part of the submission please remove all reference to it within the text.

Please clarify if there are Figure 4 and 5. Figure 5 is cited in your page 20 of the manuscript whereas the figure 4 is not mentioned. If the figure is no longer to be included as part of the submission please remove all reference to it within the text.

We are addressing both these questions together. We removed figure 5 from the manuscript to comply with one of suggested edits by our reviewers. Therefore, figures 5 and 4 do not exist for this manuscript. The mention of figure 5 was changed to Table 4 since it is referencing that correlation. Thank you for noticing this inconsistency and assisting us in fixing the situation. 

Reviewer #1: 

Line 46: the average area of urbanization was 31.0 km2. Is this area large or small. Where to compare it?

The average would be small, considering that this would be a 1/10 of the buffer size. However, you are bringing up a good point that without context this statement doesn’t provide enough substantive information for our project. We decided to remove “The average area of urbanization among the communities was 31.04 km2 ± 12.26” from the abstract. 

Lines 1111-113. Add references to your predictions.

We have added references to our predictions. 

Lines 113-115. I did not find the results from this prediction (3). Add references to your prediction. There is lot of previous publications about species richness in relation to urbanization.

For prediction three, we changed the wording to be “native species richness”

Lines 153-154. How intensively birds were counted from 20 km x 20 km squares. How equal bird species counting were in each square?

Birds were observed from 10-km radius circle within a 20-km x 20-km square. EBird Observations were collected from 2013 to 2017 within those sites. 

Lines 159-160. How closely to asymptotic species richness (xx%?). Note that the species richness increasing with bird counting intensity. What program you used to calculate rarefaction. Are rarefaction calculated each 20 km x 20 km square or larger area. Add more details.

Rarefaction curves were conducted from the information within the 10km radius. Richness of the communities were with the 95% confidence interval of the rarefaction curve. These analyses were conducted in Past 3. Additional information about rarefaction curves has been added to the manuscript. 

Lines 300 - 303. Omit 'Degrees of freedom ...'. It is twice in the table 3 text.

We omitted the degrees of freedom. 

Line 350. Is it really 'individuals'? May be it is 'species'.

Thank you. We changed individuals to species. 

Lines 351- 352. Sittidae is twice.

We have removed one of the “Sittidae” from the sentence. 

Lines 351-353. Would you add figures from those correlation.

Figures have been added to the manuscript. 

Figure 1. Would you re-draw this figure. See more details: McGeoch & Gaston 2002: Occupancy frequency distributions: patterns, artefacts and mechanisms. Biological Reviews, 77, 311-331. Write also some words to the results and discussion sections. You can also analysed the distribution pattern see more details:

Hui C. (2012) Scale effect and bimodality in the frequency distribution of species occupancy. Community Ecology, 13, 30-35.

Jenkins D.G. (2011) Ranked species occupancy curves reveal common patterns among diverse metacommunities. Global Ecology and Biogeography, 20, 486-497.

We are addressing both comments with this response. We redid our graph to include proportions of species across the sites. This graph further shows that some species are common (core), and many are rare (satellites). We added information on core and satellite species to the manuscript. We are hesitant to dive deeper into assessing core-satellite distribution, as this is not the focus of the paper, and reviewer 2 suggested to cut down the results section. 

Reviewer #2: 

Title: In my humble opinion, this study is not about metacommunities. The title is not very informative and should be modified.

We added information on how our study relates to the metacommunity concept. 

L41 “functional richness of diet” sounds odd. Please, reword.

We changed the wording to “functional richness based on diet”.

L45-46 This info is not relevant here.

The referenced sentence “Increasing urbanization was positively related to number of canopy foragers, while emergent wetlands were negatively related to species with frugivorous diets or those that nested on cliffs/buildings” was removed.

L50-51 This is a rather vague statement. Please, elaborate a bit.

We have made the statement clearer to the reader. “Spatial and environmental factors played an important role in taxonomic and functional structure in avian metacommunity structure.”

L51-53 I think the conclusions of this study should be much improved in order to attract the attention of a broad audience.

We have improved our discussion section to make it more attractive to readers. 

L111-115 Some of these predictions constitute well-known patterns and someone would argue that rather than hypotheses to be tested, they constitute truisms. Please, specify the main novelty of this study in relation to previous work.

Bird Data: It seems that bird surveys were conducted in different habitats along an environmental gradient. It is known that bird detectability can vary among habitats (e.g., detection probability is higher in open vs. closed habitats). How did you account for detection biases in this study?

EBird is a semi-structured citizen database. To decrease biases and increase detection probability, we followed strict filtering methods as done by other researchers. First off, we used complete checklists that reduce observation bias/preference toward a certain species. We also focused on metadata that include information on effort. We used checklists that included time duration, travel and sampling type. Studies have shown that including the filtering methods like the ones that we did for this study (and is addressed within the manuscript), increases the accuracy of the results. By adding these requirements, models have improved even in less sparse regions (Johnston et al. 2019). 

Since we were working with communities, we also examined the rarefaction curve for each community sampled. This helped us determine the sampling effort of each community or location. If one did not reach an asymptote, it was not included in the study. It did indicate how well-sampled each community was and if we can compare diversity. 

Johnston, A., et al. "Best practices for making reliable inferences from citizen science data: case study using eBird to estimate species distributions." BioRxiv 574392 (2019).

Trait Data: The Hand Wing index is a better proxy for dispersal capacity than wing length. This variable can be obtained from recent studies (see e.g., Sheard et al. 2020. Nature Communications).

We appreciate you recommending this manuscript. Hand wing index is a good proxy for dispersal capacity, but we did not have access to this type of data when it was being written. Unfortunately, the list from Sheard et al. 2020 has only 77% of our species. To avoid losing data, we have decided to stay with wing length, but mention in the discussion how hand wing index is a better proxy. 

Trait Data: Foraging variables (%): it is likely these variables are correlated so I think they could be summarized into 1-2 axes by means of a Principal Component Analyses without loss of information.

When conducting an RLQ analysis, all three matrices go through an ordination analysis. For functional traits, which includes foraging variables, we conducted a hill-smith ordination since there were categorical variables. We have added this information into the manuscript. 

Trait Data: Body mass (length) and bill length are highly correlated, so I would use the residuals of bill length (i.e., size-corrected bill length) after a phylogenetic size-correction (Revell, 2009) instead of the raw variable.

Most likely, morphology has phylogenetic structure. However, by removing effects of phylogeny we risk losing a lot of ecological signal as well. Therefore, we have decided not to remove phylogenetic structure in from morphological measurements. 

I wonder why authors do not use other more traditional measures of functional diversity instead of functional richness, a metric that depends on species richness.

Rao’s quadratic entropy is a commonly used index of functional richness and incorporates trait differences between species. Due to the inherit nature of eBird (e.g., surveyors not covering the entire area, the uncertainty of abundances being counted properly, etc.), we decided relative abundance was not reliable. Instead, we decided species richness was more reliable and still a great indicator of environmental and spatial processes on avian metacommunity. Like we mentioned in the manuscript, we used presence/absence for site x species matrix. When you remove species abundance from Rao’s quadratic entropy, it becomes functional richness instead. 

L262-265 This info should be given in Material and Methods.

This information has been moved to the material and methods section “Spatial and Environmental Data – R matrix”.

L265-272 This belongs to M&M.

We respectfully disagree with this idea. This paragraph are the results of the principal components analyses and sets the stage for the rest of the results. 

L305 What about the relationship between species richness and environmental variables (e.g., urbanization gradient)? Is species richness significantly associated with trait richness?

According to the RDA, as seen in figure 2c, we evaluate the relationship between development and trait and species richness. Invasive species richness had a greater association than native species richness. Native species richness, bill length, wing length, and diet were highly associated with one another, while there was a high association between invasive species richness and nest type richness. These associations are probably due to traits reacting in the same way to environmental variables. 

Results: This section is too wordy and not easy to read. I think it should be trimmed by half.

We have trimmed down the results section. 

L369-379 This paragraph fits better in Introduction.

We changed the paragraph to fit better in the discussion.

L381-384 In my opinion, these results are not novel at all. These are rather platitudes.

We changed our discussion section to connect it more to a metacommunity analysis. 

L384-385 Another quite obvious result. Obviously, trait diversity will be higher in ecosystems where terrestrial and semi-aquatic species coexist.

We did remove this observation, but we do not find it an obvious result. We know that taxonomic richness is high, but that doesn’t necessarily mean functional richness will be high in aquatic wetlands. Insectivorous birds may have a higher association with aquatic wetlands, but this ecosystem may not be conducive to other dietary guilds. 

L390-391 Indeed, trees and cliffs are scarcer in wetlands, so this relationship is quite obvious and lacking of interest.

This result was removed from the discussion section.

L404 Traits like brain mass (which is available for a large number of species) would be of interest in this context.

We agree encephalization would be interesting to study and will look into that for future studies. 

439-441 Conclusions are not conclusive at all and the take-home message is a bit disappointing. Authors should emphasize the main merits of their study.

We changed this to include a more comprehensive take-home message. 

Fig. 3b: The number of variables is so high that this figure is hardly interpretable.

We changed our figure to make it easier to read. 

Fig. 3d: Unclear figure. It is almost impossible to discern among the large number of symbols used to identify each family.

We agree this figure is hard to interpret. Since it only adds a little bit of information to the manuscript and we are trying to cut down the results section, we have decided to remove figure 3d. 

Thank you so much for your time and your valuable edits. We hope that you like the newly revised manuscript.

Best,

Erin E. Stukenholtz, M. Sc.

Ph.D. candidate

Natural Resources Management

Texas Tech University

Lubbock, TX 79414

---

## [Editor Report · Decision Letter 1]

16 May 2022

PONE-D-21-38671R1Taxonomic and functional components of avian metacommunity structurePLOS ONE

Dear Dr. Stukenholtz,

Thank you for submitting your revised manuscript to PLOS ONE. However you overlooked my comments that were given in a pdf file (my statement in the first decision was: "You will find my detailed comments in the attached file (PONE-D-21-38671_LFB.pdf)"). I checked your revised text and found out that you did not consider my suggestions (e.g., the legend of Table 2 was not corrected concerning the mistaken use of "eigenvalues"). Consequently, I am asking you to make a second revision of your manuscript.

We look forward to receiving your revised manuscript.

Kind regards,

Louis-Felix Bersier, Ph.D.

Academic Editor

PLOS ONE
---

## [Author Response · Author response to Decision Letter 1]

31 May 2022

Comments from Editor Dr. Louis-Felix Bersier:

I think that the title should be more specific. The urbanization gradient should be mentioned.

We have added urban gradient to the title. 

Necessary in the abstract?

This was removed from the abstract. 

Well, they did not combined both approaches into one, but they proposed to use both approaches in combination (as you did). Please rephrase.

Thank you for pointing this out. We changed the phrasing to be more accurate. 

I would expect some predictions (or general questions) linked to metacommunity theory highlighted above.

We have added more predictions linked to the metacommunity theory. 

Quite obscure: what is the grid used for? What do you do with the buffer?

We clarified the grid and buffer was used for in our research. 

This is astonishingly large to me for such a small bird. I checked the reference and found that it concerns birds outside the breeding season. This raises the question: was season considered in the selection of the data? This should be indicated and your choice should be justified (without selection, season may be treated as a supplementary factor?).

Yes, season was considered in this research. We focused on bird observations during summer, when species arrive to their migratory breeding grounds. 

How can a point "intersect" a buffer. The whole explanation of what is included in your "communities", and what is their spatial limits is totally obscure to me. Please better explain (a figure in the SI may be useful).

We added more information to help clarify what we did. 

How do you judge that an asymptote is reached ? What rarefaction model (and software?) did you use?

This information has been added to the manuscript.

What are "percent foraging strategies" ?

This has been clarified in the manuscript. 

Not clear. Perhaps "We coded nesting stragegy as dummy variables" (this is similar for dietary guild). Status would be a binary variable since this variable has two exclusive levels.

Thank you, we used your suggestion in the manuscript. 

Not clear. Do you mean "rather than assessing quantitative characteristics at the species level" ?

We clarified this statement in the manuscript. 

I would conclude that a "buffer" is a sampling site? Again, what you consider a "sampling site" must be clearly defined.

We defined that the buffers are our sampling sites in the manuscript. 

They should be listed in a table, idealy in the SI

Land cover types are mentioned in Table 2 with PCA loadings. We can add more information in SI if necessary. 

Reference needed

Reference added. 

Not clear... Did you compute a correlation matrix between site and then a PCoA ? In this case, this would be problematic as Pearson correlation is not a good measure of site similarity (see the book of Legendre and Legendre). If you made the PCA directely on the environmental matrix, I would remove "with a correlation matrix".

We removed “correlation matrix” to make it more clear. 

“after a significant global model,” Not clear.

That was discussing the previous step, but we removed it to avoid confusion. 

I would say "1) variation in species composition uniquely ...." and similarly for points 2) and 3)

We took this suggestion and added it to the manuscript. 

This needs more explanation.

We clarified our methods for the RDA on trait richness and spatial and environmental processes. 

This is always the case in permutation tests and this does not reflect the suggestion in reference 56. Do you mean here "Model 4" ?

We removed the sentence on permutation test and clarified the sentence about the type I error rate. 

Then you have to highlght these species in Fig. 2a. (e.g. with numbers)

This paragraph was removed to decrease the size of our results section. 

Why "taxonomy" and not "species occurrence" or "species abundance" or "community composition" ?

Taxonomy was changed to species occurrences.

why "richness" and not "traits" (or something else, perhaps "species traits" ?)

Richness was changed to traits.

I remark that the percentaga variance of the 2nd axes in c) and d) are extremely small. Do you have any explanation for this ?

The first axis for both RDA tests were significantly correlated with the variables. The second axis didn’t have a significant relationship with any of the variables. For ease, we did use a biplot to show the correlations, but did discuss that the second axis didn’t account for a significant amount of the variation. 

This part should be placed in the Method section, as it does not directly concern the bird communities (when you describe how you reduced the matrix with environmental data).

This is similar to what Reviewer 2 discussed. We do disagree because this sets up the stage for the rest of the analyses. We believe the reader will be able to follow the analyses better with these results coming right before the constrained analyses and RLQ and fourth-corner analyses. 

... and mixed forests and woody wetlands (from Table 2).

This has been added. 

Although you use the community matrix to extract the significant PCNMs, this part could also be placed in the Method section

We believe these results can help with the reader follow the analyses better. We did add more information to allow readers understand these results better. 

These values cannot be "eigenvalues". Eigenvalues are linked to principal components, and not to the "variables". I guess that these are the coordinates for the variables for the three principal components. This table could also be placed in the Method section, or

Yes, these are coordinates. Thank you for catching that mishap. We have corrected our error. 

These results are strange at first sight. How can you have very similar coordinates for low, mid, and high development? The only explanation I can think of is that these three variables have non-zero values only in urban environments. In this case, they appear very redundant and could potentially be merged into a single variable. This highlights the need to give a description of all variables of table 2.

Low, mid, and high represent three different characteristics of cities and the value of impervious surfaces. Low development is representative of residential areas with 20 to 49% impervious surfaces. Mid development represents residential areas with higher percent of impervious surfaces (50% to 79%). High development is representative of the commercial area in cities with 80% to 100% impervious surfaces. With these three categories represent the urban gradient from low to high in larger cities. However, not all cities have a strong urban gradient (e.g., cities with smaller populations). The intensity of the development can influence avian composition. Describing the different aspects of development can further assist in understanding species response to urban gradient. With this in mind, we don’t feel it would be a good idea to combine the development into one variable. 

This is not obvious from the Fig. 2a. I would add in the figure or in the legend a description of the gradient linked to each principal component of the environmental matrix.

We added PC and PCNM information to the caption. 

A problem here is that spatial variation has 5 variables (against 3 for environment), which inevitably increases the variation explained by this group. Note that it is a general issue when comparing percentages of explained

Yes, this is true. Adjusted R2 accounts for the increase in independent variables. We added this information to the manuscript. 

axis ? Instead of first gradient.

This is correct, but this section was removed from the newest draft. 

of envrironmental and spatial variables ?

Thank you, we added variables to the caption. 

This type of information can be seen from Fig. 2d. Rather than this very abstract statement, it would be useful to refer to the type of spatial gradient (eg small-scale or fine-grained, medium-scale...) that is represented by each PCNM variable. 

Also, in Figs 2a and 2b, it is not possible to see which species is associated with which PCNM variable. This is problematic since the potentially interesting biological information is totally hidden. As I said above, Tables with the coordinates are needed. You could additionally label the most "interesting" species in the figures. 

We added more information on fine to coarse spatial structures in the manuscript. For supplemental material we are added a table of species coordinates for the CCAs.

This value does not correspond to the one in Fig. 3a

This was fixed to be correct. 

Figure 3

Same remark as above concerning the interpretation of the figure (panel a)

should it be the plain lines ?

There may be interesting information here, but it is cryptic without legends in the figure.

I would mention that the Mexican Jay is placed outside the limits of the graph.

It made this figure and caption a lot clearer. We added information on PCs and PCNMs, changed the lines to colors, and removed figure 3d, since it was too complicated. 

Again, the biological information is very difficult to extract. This part is more interesting, but I would still try to explain in words the biological information (not using only RLQ axis 1 or PCNM4, but what type of variation or of relationship they express)

Thank you for helping us make our results section clearer to readers. We connected traits to the gradients behind PCs and spatial scales of PCNMs.

Although used as is in Ref. 62, I would be careful by the use of "stochasticity" as a surrogate for "unexplained variation". Unexplained variation may be due to factors not included in the models. So, I would rephrase this sentence (eg, "Similar to our study, the large percentage of unexplained variation indicates that stochasticity was...")

We added your suggestion. 

Where is Fig. 5 ?

It was changed to another figure in an earlier draft. This has been corrected. 

Our study...

Thank you, this has been corrected. 

Comments from Editor Freddie Domini:

We note that you have stated that you will provide repository information for your data at acceptance. Should your manuscript be accepted for publication, we will hold it until you provide the relevant accession numbers or DOIs necessary to access your data. If you wish to make changes to your Data Availability statement, please describe these changes in your cover letter and we will update your Data Availability statement to reflect the information you provide.

Data for this study was collected from a citizen scientist database. Therefore, we will need to change our Data Availability Statement. We will not have data to provide once accepted, since it is already publicly available and does not belong to us. Thank you for helping us to clarify this situation.

Thank you for stating the following financial disclosure:

The authors received no specific funding for this work.

Please upload a copy of Figure 5, to which you refer in your text on page 20. If the figure is no longer to be included as part of the submission please remove all reference to it within the text.

Please clarify if there are Figure 4 and 5. Figure 5 is cited in your page 20 of the manuscript whereas the figure 4 is not mentioned. If the figure is no longer to be included as part of the submission please remove all reference to it within the text.

We are addressing both these questions together. We removed figure 5 from the manuscript to comply with one of suggested edits by our reviewers. Therefore, figures 5 and 4 do not exist for this manuscript. The mention of figure 5 was changed to Table 4 since it is referencing that correlation. Thank you for noticing this inconsistency and assisting us in fixing the situation. 

Reviewer #1: 

Line 46: the average area of urbanization was 31.0 km2. Is this area large or small. Where to compare it?

The average would be small, considering that this would be a 1/10 of the buffer size. However, you are bringing up a good point that without context this statement doesn’t provide enough substantive information for our project. We decided to remove “The average area of urbanization among the communities was 31.04 km2 ± 12.26” from the abstract. 

Lines 1111-113. Add references to your predictions.

We have added references to our predictions. 

Lines 113-115. I did not find the results from this prediction (3). Add references to your prediction. There is lot of previous publications about species richness in relation to urbanization.

For prediction three, we changed the wording to be “native species richness”

Lines 153-154. How intensively birds were counted from 20 km x 20 km squares. How equal bird species counting were in each square?

Birds were observed from 10-km radius circle within a 20-km x 20-km square. EBird Observations were collected from 2013 to 2017 within those sites. 

Lines 159-160. How closely to asymptotic species richness (xx%?). Note that the species richness increasing with bird counting intensity. What program you used to calculate rarefaction. Are rarefaction calculated each 20 km x 20 km square or larger area. Add more details.

Rarefaction curves were conducted from the information within the 10km radius. Richness of the communities were with the 95% confidence interval of the rarefaction curve. These analyses were conducted in Past 3. Additional information about rarefaction curves has been added to the manuscript. 

Lines 300 - 303. Omit 'Degrees of freedom ...'. It is twice in the table 3 text.

We omitted the degrees of freedom. 

Line 350. Is it really 'individuals'? May be it is 'species'.

Thank you. We changed individuals to species. 

Lines 351- 352. Sittidae is twice.

We have removed one of the “Sittidae” from the sentence. 

Lines 351-353. Would you add figures from those correlation.

Figures have been added to the manuscript. 

Figure 1. Would you re-draw this figure. See more details: McGeoch & Gaston 2002: Occupancy frequency distributions: patterns, artefacts and mechanisms. Biological Reviews, 77, 311-331. Write also some words to the results and discussion sections. You can also analysed the distribution pattern see more details:

Hui C. (2012) Scale effect and bimodality in the frequency distribution of species occupancy. Community Ecology, 13, 30-35.

Jenkins D.G. (2011) Ranked species occupancy curves reveal common patterns among diverse metacommunities. Global Ecology and Biogeography, 20, 486-497.

We are addressing both comments with this response. We redid our graph to include proportions of species across the sites. This graph further shows that some species are common (core), and many are rare (satellites). We added information on core and satellite species to the manuscript. We are hesitant to dive deeper into assessing core-satellite distribution, as this is not the focus of the paper, and reviewer 2 suggested to cut down the results section. 

Reviewer #2: 

Title: In my humble opinion, this study is not about metacommunities. The title is not very informative and should be modified.

We added information on how our study relates to the metacommunity concept. 

L41 “functional richness of diet” sounds odd. Please, reword.

We changed the wording to “functional richness based on diet”.

L45-46 This info is not relevant here.

The referenced sentence “Increasing urbanization was positively related to number of canopy foragers, while emergent wetlands were negatively related to species with frugivorous diets or those that nested on cliffs/buildings” was removed.

L50-51 This is a rather vague statement. Please, elaborate a bit.

We have made the statement clearer to the reader. “Spatial and environmental factors played an important role in taxonomic and functional structure in avian metacommunity structure.”

L51-53 I think the conclusions of this study should be much improved in order to attract the attention of a broad audience.

We have improved our discussion section to make it more attractive to readers. 

L111-115 Some of these predictions constitute well-known patterns and someone would argue that rather than hypotheses to be tested, they constitute truisms. Please, specify the main novelty of this study in relation to previous work.

Bird Data: It seems that bird surveys were conducted in different habitats along an environmental gradient. It is known that bird detectability can vary among habitats (e.g., detection probability is higher in open vs. closed habitats). How did you account for detection biases in this study?

EBird is a semi-structured citizen database. To decrease biases and increase detection probability, we followed strict filtering methods as done by other researchers. First off, we used complete checklists that reduce observation bias/preference toward a certain species. We also focused on metadata that include information on effort. We used checklists that included time duration, travel and sampling type. Studies have shown that including the filtering methods like the ones that we did for this study (and is addressed within the manuscript), increases the accuracy of the results. By adding these requirements, models have improved even in less sparse regions (Johnston et al. 2019). 

Since we were working with communities, we also examined the rarefaction curve for each community sampled. This helped us determine the sampling effort of each community or location. If one did not reach an asymptote, it was not included in the study. It did indicate how well-sampled each community was and if we can compare diversity. 

Johnston, A., et al. "Best practices for making reliable inferences from citizen science data: case study using eBird to estimate species distributions." BioRxiv 574392 (2019).

Trait Data: The Hand Wing index is a better proxy for dispersal capacity than wing length. This variable can be obtained from recent studies (see e.g., Sheard et al. 2020. Nature Communications).

We appreciate you recommending this manuscript. Hand wing index is a good proxy for dispersal capacity, but we did not have access to this type of data when it was being written. Unfortunately, the list from Sheard et al. 2020 has only 77% of our species. To avoid losing data, we have decided to stay with wing length, but mention in the discussion how hand wing index is a better proxy. 

Trait Data: Foraging variables (%): it is likely these variables are correlated so I think they could be summarized into 1-2 axes by means of a Principal Component Analyses without loss of information.

When conducting an RLQ analysis, all three matrices go through an ordination analysis. For functional traits, which includes foraging variables, we conducted a hill-smith ordination since there were categorical variables. We have added this information into the manuscript. 

Trait Data: Body mass (length) and bill length are highly correlated, so I would use the residuals of bill length (i.e., size-corrected bill length) after a phylogenetic size-correction (Revell, 2009) instead of the raw variable.

Most likely, morphology has phylogenetic structure. However, by removing effects of phylogeny we risk losing a lot of ecological signal as well. Therefore, we have decided not to remove phylogenetic structure in from morphological measurements. 

I wonder why authors do not use other more traditional measures of functional diversity instead of functional richness, a metric that depends on species richness.

Rao’s quadratic entropy is a commonly used index of functional richness and incorporates trait differences between species. Due to the inherit nature of eBird (e.g., surveyors not covering the entire area, the uncertainty of abundances being counted properly, etc.), we decided relative abundance was not reliable. Instead, we decided species richness was more reliable and still a great indicator of environmental and spatial processes on avian metacommunity. Like we mentioned in the manuscript, we used presence/absence for site x species matrix. When you remove species abundance from Rao’s quadratic entropy, it becomes functional richness instead. 

L262-265 This info should be given in Material and Methods.

This information has been moved to the material and methods section “Spatial and Environmental Data – R matrix”.

L265-272 This belongs to M&M.

We respectfully disagree with this idea. This paragraph are the results of the principal components analyses and sets the stage for the rest of the results. 

L305 What about the relationship between species richness and environmental variables (e.g., urbanization gradient)? Is species richness significantly associated with trait richness?

According to the RDA, as seen in figure 2c, we evaluate the relationship between development and trait and species richness. Invasive species richness had a greater association than native species richness. Native species richness, bill length, wing length, and diet were highly associated with one another, while there was a high association between invasive species richness and nest type richness. These associations are probably due to traits reacting in the same way to environmental variables. 

Results: This section is too wordy and not easy to read. I think it should be trimmed by half.

We have trimmed down the results section. 

L369-379 This paragraph fits better in Introduction.

We changed the paragraph to fit better in the discussion.

L381-384 In my opinion, these results are not novel at all. These are rather platitudes.

We changed our discussion section to connect it more to a metacommunity analysis. 

L384-385 Another quite obvious result. Obviously, trait diversity will be higher in ecosystems where terrestrial and semi-aquatic species coexist.

We did remove this observation, but we do not find it an obvious result. We know that taxonomic richness is high, but that doesn’t necessarily mean functional richness will be high in aquatic wetlands. Insectivorous birds may have a higher association with aquatic wetlands, but this ecosystem may not be conducive to other dietary guilds. 

L390-391 Indeed, trees and cliffs are scarcer in wetlands, so this relationship is quite obvious and lacking of interest.

This result was removed from the discussion section.

L404 Traits like brain mass (which is available for a large number of species) would be of interest in this context.

We agree encephalization would be interesting to study and will look into that for future studies. 

439-441 Conclusions are not conclusive at all and the take-home message is a bit disappointing. Authors should emphasize the main merits of their study.

We changed this to include a more comprehensive take-home message. 

Fig. 3b: The number of variables is so high that this figure is hardly interpretable.

We changed our figure to make it easier to read. 

Fig. 3d: Unclear figure. It is almost impossible to discern among the large number of symbols used to identify each family.

We agree this figure is hard to interpret. Since it only adds a little bit of information to the manuscript and we are trying to cut down the results section, we have decided to remove figure 3d. 

Thank you so much for your time and your valuable edits. We hope that you like the newly revised manuscript.

Best,

Erin E. Stukenholtz, M. Sc.

Ph.D. candidate

Natural Resources Management

Texas Tech University

Lubbock, TX 79414

---

## [Editor Report · Decision Letter 2]

30 Jun 2022

Taxonomic and functional components of avian metacommunity structure along an urban gradient

PONE-D-21-38671R2

Dear Dr. Stukenholtz,

We’re pleased to inform you that your manuscript has been judged scientifically suitable for publication and will be formally accepted for publication once it meets all outstanding technical requirements.

Kind regards,

Louis-Felix Bersier, Ph.D.

Academic Editor

PLOS ONE

Additional Editor Comments (optional):

I read carefully your second revision with regard to the remarks of both Reviewers and to my own comments. I found some minor editorial issues that you will find in the attached pdf file (PONE-D-21-38671_R2_LFB.pdf). Please consider them carefully for your final manuscript.

I want to congratulate you for your thorough corrections. Your text is now much clearer and I had no problem in grasping the methods and results. Also, the Discussion section is now biologically much more interesting. I all, it is a very nice and useful contribution to community ecology with top-of-the-line analytical tools.

---

## [Editor Report · Acceptance letter]

19 Jul 2022

PONE-D-21-38671R2 

Taxonomic and functional components of avian metacommunity structure along an urban gradient

Dear Dr. Stukenholtz:

I'm pleased to inform you that your manuscript has been deemed suitable for publication in PLOS ONE. Congratulations! Your manuscript is now with our production department. 

Kind regards, 

on behalf of

Prof Louis-Felix Bersier 

Academic Editor

PLOS ONE